DOI: 10.1038/s41467-018-07466-6　　OPEN

# Comprehensive human cell-type methylation atlas reveals origins of circulating cell-free DNA in health and disease

Joshua Moss[1,2], Judith Magenheim[1], Daniel Neiman[1], Hai Zemmour [1], Netanel Loyfer[2], Amit Korach[3], Yaacov Samet[4], Myriam Maoz[5], Henrik Druid [6,7], Peter Arner[8], Keng-Yeh Fu[9], Endre Kiss[9], Kirsty L. Spalding[8,9], Giora Landesberg[10], Aviad Zick [5], Albert Grinshpun[5], A.M.James Shapiro[11], Markus Grompe [12], Avigail Dreazan Wittenberg[1], Benjamin Glaser [13], Ruth Shemer[1], Tommy Kaplan [2] & Yuval Dor[1]

Methylation patterns of circulating cell-free DNA (cfDNA) contain rich information about recent cell death events in the body. Here, we present an approach for unbiased determination of the tissue origins of cfDNA, using a reference methylation atlas of 25 human tissues and cell types. The method is validated using in silico simulations as well as in vitro mixes of DNA from different tissue sources at known proportions. We show that plasma cfDNA of healthy donors originates from white blood cells (55%), erythrocyte progenitors (30%), vascular endothelial cells (10%) and hepatocytes (1%). Deconvolution of cfDNA from patients reveals tissue contributions that agree with clinical findings in sepsis, islet transplantation, cancer of the colon, lung, breast and prostate, and cancer of unknown primary. We propose a procedure which can be easily adapted to study the cellular contributors to cfDNA in many settings, opening a broad window into healthy and pathologic human tissue dynamics.

[1] Department of Developmental Biology and Cancer Research, Institute for Medical Research Israel-Canada, The Hebrew University-Hadassah Medical School, Jerusalem 9112001, Israel. [2] School of Computer Science and Engineering, The Hebrew University of Jerusalem, Jerusalem 9190401, Israel. [3] Department of Cardio-Thoracic Surgery, Hadassah-Hebrew University Medical Center, Jerusalem 9112001, Israel. [4] Department of Vascular Surgery, Hadassah-Hebrew University Medical Center, Jerusalem 9112001, Israel. [5] Department of Oncology, Hadassah-Hebrew University Medical Center, Jerusalem 9112001, Israel. [6] Department of Oncology-Pathology, Karolinska Institutet, SE17177 Stockholm, Sweden. [7] Dept of Forensic Medicine, The National Board of Forensic Medicine, SE11120 Stockholm, Sweden. [8] Department of Medicine, Karolinska University Hospital, Karolinska Institutet, SE17176 Stockholm, Sweden. [9] Department of Cell and Molecular Biology, Karolinska Institutet, SE17177 Stockholm, Sweden. [10] Dept of Anesthesiology and Critical Care Medicine, Hadassah-Hebrew University Medical Center, 9112001 Jerusalem, Israel. [11] Department of Surgery and the Clinical Islet Transplant Program, University of Alberta, Edmonton, AB T6G 2R3, Canada. [12] Papé Family Pediatric Research Institute, Oregon Health & Science University, Portland, OR 97239, USA. [13] Dept of Endocrinology and Metabolism Service, Hadassah-Hebrew University Medical Center, 9112001 Jerusalem, Israel. Correspondence and requests for materials should be addressed to R.S. (email: shemer.ru@mail.huji.ac.il) or to T.K. (email: tommy@cs.huji.ac.il) or to Y.D. (email: yuvald@ekmd.huji.ac.il)

Small fragments of DNA circulate freely in the peripheral blood of healthy and diseased individuals. These cell-free DNA (cfDNA) molecules are thought to originate from dying cells and thus reflect ongoing cell death taking place in the body[1]. In recent years, this understanding has led to the emergence of diagnostic tools, which are impacting multiple areas of medicine. Specifically, next-generation sequencing of fetal DNA circulating in maternal blood has allowed non-invasive prenatal testing (NIPT) of fetal chromosomal abnormalities[2,3]; detection of donor-derived DNA in the circulation of organ transplant recipients can be used for early identification of graft rejection[4,5]; and the evaluation of mutated DNA in circulation can be used to detect, genotype and monitor cancer[1,6]. These technologies are powerful at identifying genetic anomalies in circulating DNA, yet are not informative when cfDNA does not carry mutations.

A key limitation is that sequencing does not reveal the tissue origins of cfDNA, precluding the identification of tissue-specific cell death. The latter is critical in many settings such as neuro-degenerative, inflammatory or ischemic diseases, not involving DNA mutations. Even in oncology, it is often important to determine the tissue origin of the tumor in addition to determining its mutational profile, for example in cancers of unknown primary (CUP) and in the setting of early cancer diagnosis[7]. Identification of the tissue origins of cfDNA may also provide insights into collateral tissue damage (e.g., toxicity of drugs in genetically normal tissues), a key element in drug development and monitoring of treatment response.

Several approaches have been proposed for tracing the tissue sources of cfDNA, based on tissue-specific epigenetic signatures. Snyder et al. have used information on nucleosome positioning in various tissues to infer the origins of cfDNA, based on the idea that nucleosome-free DNA is more likely to be degraded upon cell death and hence will be under-represented in cfDNA[8]. Ulz et al. used this concept to infer gene expression in the cells contributing to cfDNA[9]. The latter can theoretically indicate not only the tissue origins of cfDNA, but also cellular states at the time of cell death, for example whether cells died and released cfDNA while engaged in the cell division cycle or during quiescence.

An alternative approach is based on DNA methylation patterns. Methylation of cytosine adjacent to guanine (CpG sites) is an essential component of cell type-specific gene regulation, and hence is a fundamental mark of cell identity[10]. We and others have recently shown that cfDNA molecules from loci carrying tissue-specific methylation can be used to identify cell death in a specific tissue[11–18]. Others have taken a genome-wide approach to the problem, and used the plasma methylome to assess the origins of cfDNA. Sun et al. inferred the relative contributions of four different tissues, using deconvolution of cfDNA methylation profiles from low-depth whole genome bisulfite sequencing (WGBS)[19]. Guo et al. demonstrated the potential of cfDNA methylation for detecting cancer as well as identifying its tissue of origin in two cancer types, using a reduced representation bisulfite sequencing (RRBS) approach[20]. Kang et al. and Li et al. described CancerLocator[21] and CancerDetector[22], probabilistic approaches for cancer detection based on cfDNA methylation sequencing.

While these studies show the potential of DNA methylation in identifying the cellular contributions to cfDNA, it remains to be seen whether cfDNA methylation can be analyzed in an unbiased and comprehensive manner, in settings where it is unclear which cell types contribute to cfDNA and which underlying diseases a patient may have. To address this challenge, we took advantage of the Illumina Infinium methylation array, which allows the simultaneous analysis of the methylation status of >450,000 CpG sites throughout the human genome. Illumina methylation arrays

have been previously used in the deconvolution of whole blood methylation profiles to determine the relative proportions of white blood cells in a sample, a crucial step in Epigenome-Wide Association Studies (EWAS)[23–25]. However, to date, array deconvolution has been applied only to whole blood samples, where all contributing cells are well-studied types of white blood cells[23].

Here we demonstrate that plasma methylation patterns can be used to accurately identify cell type-specific cfDNA in healthy and pathological conditions. We have generated an extensive reference atlas of 25 human tissues and cell types, covering major organs and cells involved in common diseases. As we show, our approach allows for a robust and accurate deconvolution of plasma methylation from as little as 20 ml of blood, and using only a small number (4039) of selected genomic loci. We quantify the major cell types contributing to cfDNA in healthy individuals, and demonstrate the origins of cfDNA in islet transplantation, sepsis and cancer. We propose principles for effective plasma methylome deconvolution, including the key importance of a reference atlas consisting of cell type, rather than whole-tissue methylomes, and discuss the potential of global cfDNA methylation analysis as a diagnostic modality for early detection and monitoring of disease.

## Results

**Development of a DNA methylation atlas**. To obtain a comprehensive DNA methylation database of human cell types, we took advantage of datasets which were previously published, either as part of The Cancer Genome Atlas (TCGA)[26] or by individual groups that deposited data in the Gene Expression Omnibus (GEO). In selecting datasets to be included in the database, we used the following criteria: (1) we only used primary tissue sources, which have not been passaged in culture—reasoning that culture may change methylation patterns or alter the cellular composition of a mixed tissue, e.g., enriched for fibroblasts; (2) used the methylomes of healthy human tissues, which are expected to be universally conserved (that is, be nearly identical among cells of the same type, among individuals, throughout life, and be largely retained even in pathologies)[27]; (3) excluded tissue methylomes that contained a high proportion of blood-derived DNA, as previously described[28]; (4) merged the methylomes of highly similar tissues (e.g., rectum and colon, stomach and esophagus, cervix and uterus); and (5) preferred the methylomes of specific cell types, rather than whole tissues. We reasoned that since whole tissues are a composite of multiple heterogeneous cell types (e.g., different types of epithelial cells, blood, vasculature and fibroblasts), methylation signatures of minority populations might be difficult to identify, and unique tissue signatures might be masked by the methylome of stroma. Unfortunately, other than isolated blood cell types, the vast majority of publically available methylomes comes from bulk tissues. We therefore generated methylation profiles of key human cell types, which have not been previously published. We have isolated primary human adipocytes, cortical neurons, hepatocytes, lung alveolar cells, pancreatic beta cells, pancreatic acinar cells, pancreatic duct cells, vascular endothelial cells and colon epithelial cells. As detailed in the Methods and Supplementary Data 1, surgical samples from each tissue were enzymatically dissociated, stained with antibodies against a cell type of interest, and isolated using either flow cytometry (FACS) or magnetic beads (MACS). We then prepared DNA from sorted cells, and obtained the genome-wide methylome using Illumina 450K or EPIC BeadChip array platforms. The result of this effort was a comprehensive human methylome reference atlas, composed of 25 tissues or cell types (Fig. 1a).

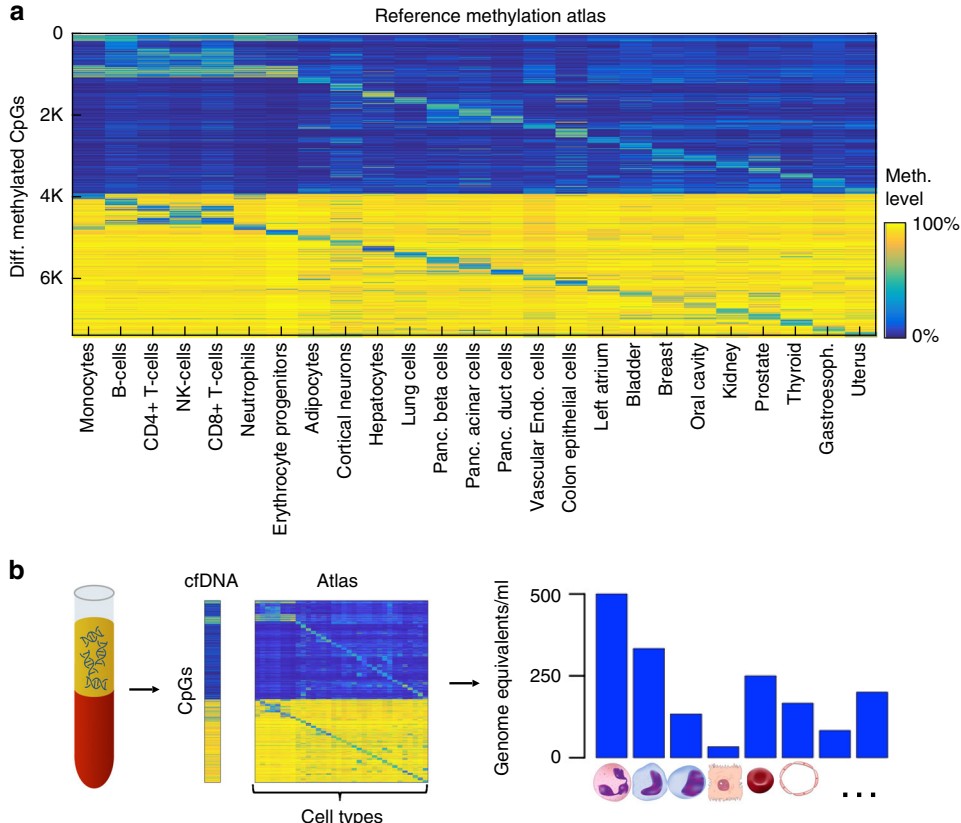

Fig. 1 Identification of tissue-of-origin of cfDNA using deconvolution of the plasma methylome aided by a comprehensive methylation atlas. **a** Methylation atlas composed of 25 tissues and cell types (columns) across ~8000 CpGs (rows). For each cell type, we selected the top 100 uniquely hypermethylated (top) and 100 most hypomethylated (bottom) CpG sites, giving a total of 5000 tissue-discriminating individual CpGs. We then added neighboring (up to 50 bp) CpGs, as well as 500 CpGs that are differentially methylated across pairs of otherwise similar tissues. Overall, we used 7890 CpGs that are located in 4039 500 bp genomic blocks. **b** Deconvolution of plasma DNA. Cell-free DNA (cfDNA) is extracted from plasma and analyzed with a methylation array. It is then deconvoluted using a reference methylation atlas to quantify the contribution of each cell type to the cfDNA sample

**Deconvolution algorithm using cell type-specific CpGs.** To analyze novel DNA methylation samples, composed of admixed methylomes from various cell types, we devised a computational deconvolution algorithm. We approximate the plasma cfDNA methylation profile as a linear combination of the methylation profiles of cell types in the reference atlas. According to this model, the relative contributions of different cell types to plasma cfDNA can be determined using non-negative least squares linear regression (NNLS)[23,29,30]. In addition, the relative contributions of cfDNA can be multiplied by the total concentration of cfDNA in plasma to obtain the absolute concentrations of cfDNA originating from each cell type (genome equivalents/ml) (Fig. 1b).

For accurate inference, we first selected a subset of CpG sites in the genome that are differentially methylated among the cell types and tissues in our atlas. We chose to use only a subset of the methylome for deconvolution based on several considerations. First, almost half of the CpG sites represented in the Illumina arrays show similar methylation patterns across all cells and are therefore uninformative. Second, we found that using a limited subset of CpGs, that are uniquely methylated or unmethylated in a cell type, allows one to detect rare cell types contributing only small amounts of cfDNA and reduces false detection of contributors (Supplementary Figures 1, 2). Third, a smaller subset of genomic regions can be the basis of a simpler, capture-based method, increasing the feasibility of routine use.

After removing CpG sites with little variance across cell types, we selected, for each tissue or cell type in the atlas, 100 CpG sites uniquely hypermethylated and 100 sites uniquely hypomethylated when compared to other tissues, as well as CpGs located adjacently (within 50 bp) to the originally selected set (Methods, Supplementary Data 1). This process resulted in ~7390 CpGs, to which we added 500 CpGs, by iteratively identifying the two most similar cell types in the atlas, and adding the CpG site upon which these two cell types differ the most (Methods, Supplementary Data 1). In total, our selection includes ~8000 CpGs, covering ~4000 genomic regions. We found this set of CpGs to perform favorably on simulated datasets when compared to other selection criteria, including the full set of CpGs (Supplementary Figures 1, 2).

**In silico mix-in simulations.** We initially performed in silico experiments to assess the performance of the deconvolution approach in determining the relative contributions of various cell types to a methylation profile of DNA from a heterogeneous mixture of cell types. For an exhaustive and realistic assessment, we used whole-blood samples from 18 individuals measured using EPIC Illumina arrays[31]. We then computationally mixed-in methylation profiles of individual samples of cell types and tissues at varying admixtures, reapplied the feature selection and deconvolution algorithms using an atlas from which the individual mixed-in sample was removed. We then compared the actual percentage with the predicted one. We simulated such data for every cell type in the reference methylation atlas, except for white blood cells, at mixing levels varying from 0 to 10% (in 1%

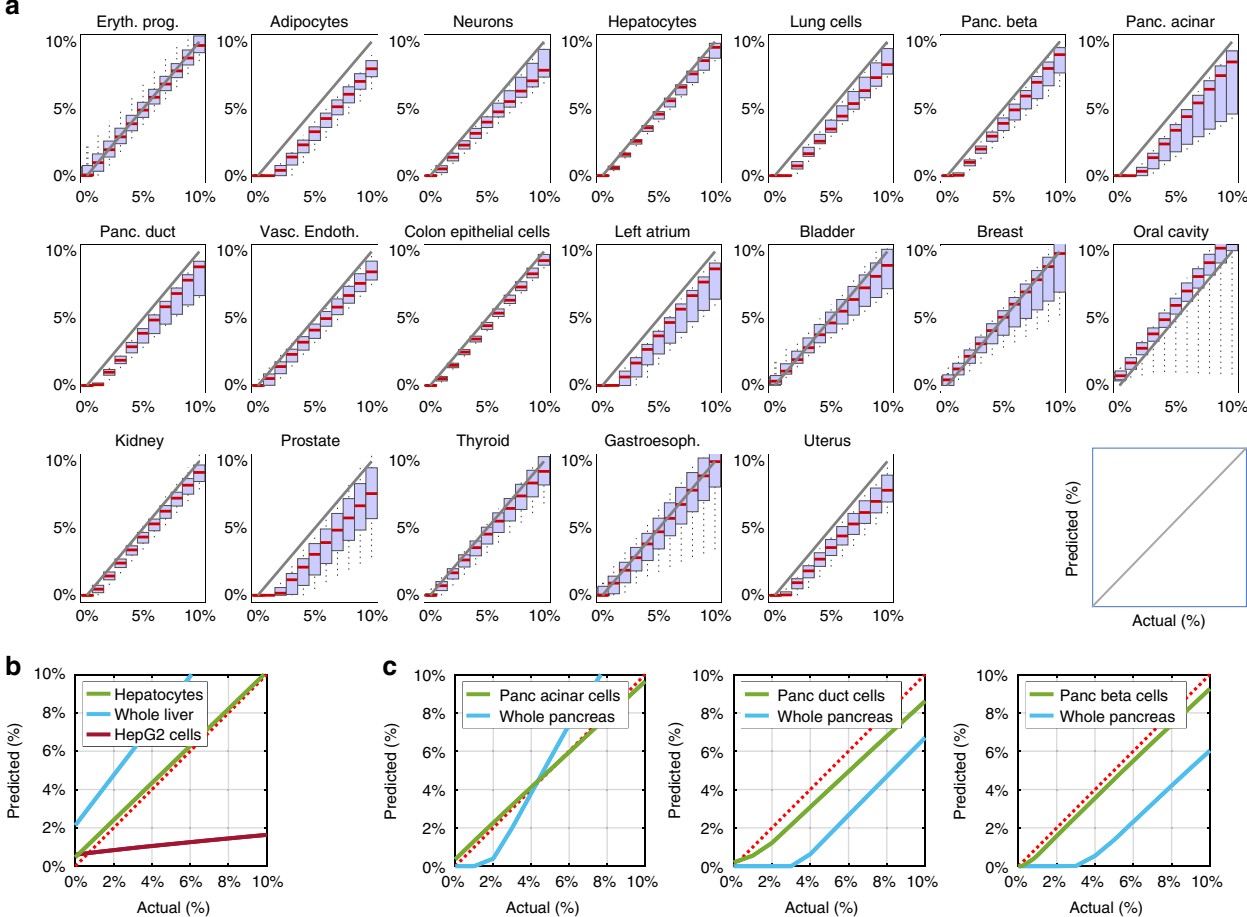

**Fig. 2** DNA methylation patterns allow for accurate deconvolution of simulated admixed samples. **a** The methylome of each cell type was mixed in silico with the methylome of leukocytes such that it contributed between 0 and 10% of DNA, in 1% intervals (x-axis of each plot) and compared to the prediction of deconvolution using our reference methylation atlas (y-axis). Red horizontal bars represent the median predicted contribution for each mixed-in level, across 36–180 replicates for each cell type (2–10 replicates of measured cell type methylomes, each mixed within any of 18 leukocyte replicates). The blue area represents a box plot spanning the 25th to 75th percentiles for each mixing ratio, with black vertical lines marking the 9th to 91st percentiles.
**b** Primary tissue methylome allows a more accurate deconvolution than whole-tissue or a cell line. Hepatocyte methylome was mixed in silico with blood methylome as in **a**. The level of inferred admixture (y-axis) was calculated using a reference tissue methylome atlas that included other hepatocyte samples (green), whole liver methylomes (blue) or the methylome of the HepG2 cell line (red). Dotted red line marks accurate prediction. **c** Cell type-specific methylomes allow a more accurate deconvolution than whole tissue methylomes. The methylome of pancreatic acinar, duct, or beta cells was diluted in silico into leukocyte methylomes (left, middle, and right, respectively); the level of admixture was calculated using a comprehensive reference atlas that contained either independent samples of the spiked-in pancreas cell types (green lines), or a whole pancreas methylome (blue lines). Note assay linearity, but reduced sensitivity, when using a whole pancreas methylome

intervals) across 36–180 replicates (18 independent leukocyte samples, times 2–10 replicates for each cell type). As shown in Fig. 2a, the deconvolution algorithm performed well for almost all cell types. Most cell types were accurately detected when composing >1% of the mixture, with many cell types detected even below 1% (Supplementary Figure 1).

Importantly, almost no non-leukocyte cells (<0.25%) were detected at mixing level of 0% (namely, analysis of pure leukocytes) (Fig. 2a, leftmost side of each plot; Supplementary Figure 1). In preliminary analysis we noticed that some confusion might occur between cell types of similar developmental origin (e.g., cervix/uterus, stomach/esophagus, colon/rectum), and therefore have merged these samples in the reference atlas (Methods). Overall the confusion between cell types was minimal, as shown using confusion matrices (Supplementary Figures 3, 4).

**Cell-type vs whole-tissue reference methylomes.** We then tested the importance of using cell type-specific versus tissue-specific or

cell-line-derived methylomes. A reference methylation atlas containing the methylome of purified hepatocytes outperformed atlases containing either whole liver or HepG2 hepatoma cell line methylomes, with the former leading to overestimation of hepatocyte in the mixture, and the latter leading to a gross underestimation (Fig. 2b). Similarly, an atlas containing the methylomes of purified pancreatic cells (acinar, duct and beta cells) was superior in detecting pancreatic DNA within blood, compared to a reference atlas containing the methylome of the whole pancreas, with the latter being ineffective in detecting small contributions (<2%) of pancreatic DNA (Fig. 2c). These findings support the feasibility of highly sensitive deconvolution of the plasma methylome, and highlight the importance of using a comprehensive, cell type-specific DNA methylation atlas for sensitive detection of rare contributors to mixed methylomes.

**In vitro DNA mixing.** We then mixed DNA samples from four specific tissues (Liver, Lung, Neurons and Colon, each from a

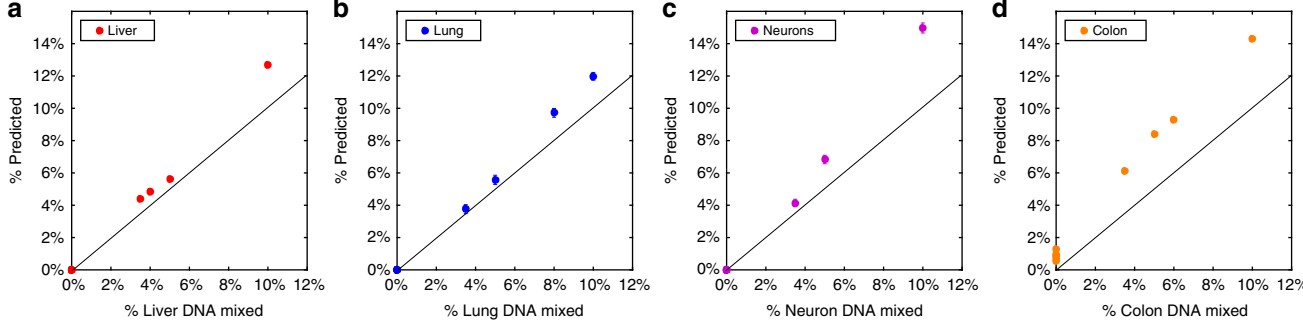

**Fig. 3** In vitro mixing experiments. Genomic DNA derived from liver (**a**), lung (**b**), neurons (**c**), and colon (**d**) (each from a single donor) was mixed in nine different combinations (detailed in Supplementary Data 1) with genomic DNA extracted from the blood of a single healthy donor, in the proportions indicated in the $X$ axis. A total of 250 ng DNA from each mixture was subjected to an Illumina EPIC array, and the resulting methylome was deconvoluted to predict the contribution of each mixed-in tissue ($Y$ axis). Each dilution point represents one mixing experiments

single donor), into leukocyte DNA from a healthy donor, at different proportions varying from 0 to 10%, and reapplied the computational deconvolution analysis (Fig. 3, Supplementary Data 1). For all samples, our algorithm identified the correct cell type in a specific and sensitive manner (Pearson's $r$ 0.88–0.99, $p$-value < 1e−3 for all mixes). These findings lend further support to the feasibility of deconvolution, but they do not fully address real-life issues such as inter-individual variation in methylation.

**Tissue origins of healthy cfDNA**. To determine the main contributors to cfDNA in healthy individuals, we collected plasma from multiple healthy donors ($n = 105$). The samples were classified by sex and age (young: 19–30 or old: 67–97; see Supplemental File 1), and cfDNA was pooled accordingly to obtain 250 ng cfDNA in each pool.

We then obtained methylation profiles of each sample ($n = 8$) using Illumina arrays and performed a deconvolution analysis to estimate the relative contribution of each tissue/cell-type to the cfDNA. The predicted distribution of contributing tissues/cell types was similar among all pools (Fig. 4a, b). Additionally, cfDNA from four additional healthy individuals was analyzed and found to be consistent with the findings in the pooled samples (Supplementary Data 1). As previously reported[32], we found that the main contributors to cfDNA were of hematopoietic origin. On average, 32.0% (±1.1% mean SD) of cfDNA came from granulocytes, 29.7% (±0.8%) from erythrocyte progenitors, 10.5% (±1.1%) from monocytes, and 12.1% (±0.7%) from lymphocytes. The main solid tissue sources of cfDNA were vascular endothelial cells (8.6 ± 0.9%) and hepatocytes (1.2 ± 0.4%). The signal from erythrocyte progenitors, endothelial cells and hepatocytes is expected to be present in cfDNA but not in DNA isolated from leukocytes. Indeed, deconvolution of blood cell (leukocyte) methylomes predicted signals from these tissues at much lower levels than in plasma, supporting validity of the algorithm ($p < 1e−10$, Fig. 4c).

Furthermore, the predicted proportions of monocytes, neutrophils and lymphocytes in whole blood methylomes were in excellent agreement with the actual proportions of these cell types in each individual blood sample, as obtained from a complete blood count (CBC) (Fig. 4d).

Unexpectedly, deconvolution of the healthy plasma methylome revealed also a signal from neurons, accounting for as much as 2% of cfDNA (Fig. 4a, b). The significance of this finding remains to be determined, as it is not consistent with findings using PCR-sequencing of specific brain markers[11]; we favor the idea that the neuronal signal is an artifact of the assay, perhaps reflecting

contribution from a tissue not included in our atlas (see Discussion).

While the young and old samples showed similar relative contributions of the different cell types, the plasma of older people showed a significantly higher levels of total cfDNA, as measured in genome equivalents per ml of plasma (Supplementary Figure 5). The similar proportions of cfDNA origins may suggest a slower clearance rate of circulating DNA in older individuals (Fig. 4b), rather than an increased rate of cell death in all tissues. Further work is required to define the determinants of cfDNA clearance in difference physiologic and pathologic conditions. In summary, these findings provide the first detailed description of the composition of cfDNA in healthy people.

**Deconvolution of cfDNA in islet transplant recipients**. We analyzed the plasma methylome of patients with long standing type 1 diabetes, 1 h after receiving a cadaveric pancreatic islet transplant (pool of $n = 5$ recipients). The total concentration of cfDNA in these samples was ~20-fold higher than healthy control levels, suggesting a massive process of cell death shortly after islet transplantation. The deconvolution algorithm identified a large proportion (~20%) of cfDNA as derived from pancreatic origin (from beta, acinar and duct cells, Fig. 5a, b), in stark contrast to cfDNA from healthy plasma. These findings strongly support the validity of our deconvolution procedure. Strikingly, we observed that most of the increase in cfDNA levels in islet transplant recipients was of an immune cell origin (granulocytes, monocytes and lymphocytes). This finding suggests an acute immune response to the infusion of islets into recipient blood, or alternatively a response to the procedure itself and/or pre-transplant immune suppression treatment, resulting in massive immune cell death (Fig. 5b). Follow-up studies will attempt to distinguish between these possibilities.

To examine the dynamics of cfDNA of pancreatic origin, we determined the plasma methylome of three individual recipients before (<1 day), 1 h after, and 2 h after transplantation. As expected, the algorithm identified no pancreas cfDNA before islet transplantation, a large increase immediately after transplantation, and a subsequent decrease in levels of pancreatic cfDNA (Fig. 5c). Interestingly, cfDNA originating from immune cells as inferred by deconvolution showed a different dynamics, likely reflecting the response of the innate immune system to the transplantation (Supplementary Figure 6). In addition, we used a previously described targeted bisulfite-sequencing approach to quantify the amount of unmethylated CpGs at a haplotype block located over the insulin promoter[11]. We observed a high correlation ($r = 0.995$, $p ≤ 2.6e−8$) between the amount of beta

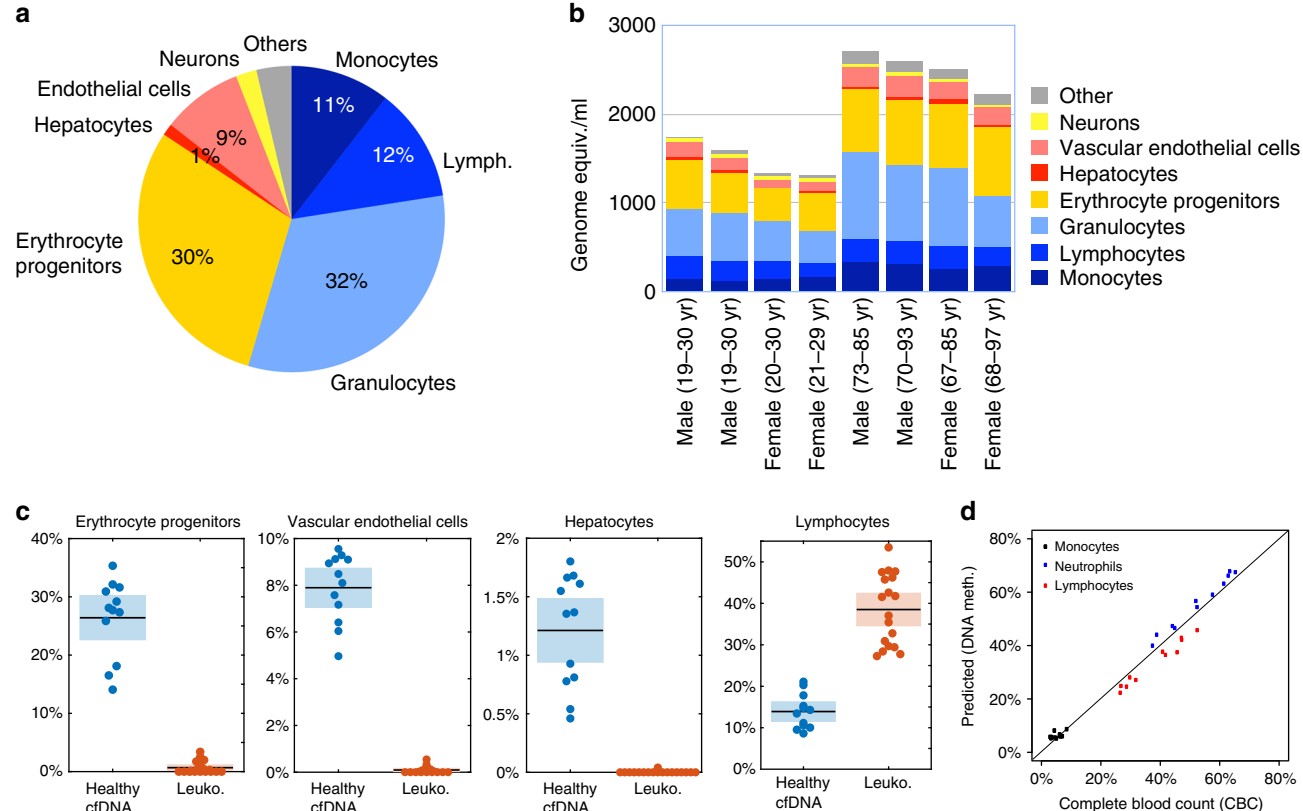

**Fig. 4** Cellular contributors to cfDNA in healthy individuals. **a** Predicted distributions of contributors to circulating cfDNA, averaged across eight sample pools of healthy donors. Contributions smaller than 1% were included in "Other". **b** Deconvolution results for eight sets of pooled DNA samples, expressed as absolute levels of DNA (genome equivalents/ml plasma, derived by multiplying the fraction contribution of each tissue by the total amount of cfDNA in 1 ml plasma). Shown are contributions larger than 1%. Young, 19–30 years old; Old, 67–97 years old (pool average > 75 yr). **c** Comparison of estimated proportion of various cell types in healthy plasma samples (blue) vs. leukocytes (orange), as predicted by deconvolution. Shown, from left, are the contributions of erythrocyte progenitor cells, vascular endothelial cells and hepatocytes, all of which are not expected in leukocyte samples. Also shown are the predicted contributions of lymphocytes, that represent a large fraction of leukocyte cell population. Shaded boxes mark 95% confidence interval of the sample mean. **d** Deconvolution of whole blood methylomes (not plasma), showing excellent correlation (Pearson's $r = 0.985$, $p < 2e−16$) between the estimated proportions of monocytes, neutrophils and lymphocytes and the actual proportions of these cells obtained via standard complete blood count (CBC) for each sample

cell cfDNA estimated by deconvolution and by targeted PCR-based method, further supporting validity of the deconvolution algorithm (Fig. 5d). Finally, we tested the deconvolution algorithm using a reference matrix containing either whole-tissue or cell type-specific methylomes. Consistently with results from deconvolution of in silico mixes (Fig. 2b, c), a reference matrix containing cell type-specific methylomes showed higher sensitivity compared with an atlas containing a whole-tissue methylome, which failed to identify pancreatic cfDNA in one of the three recipients (Fig. 5e).

**The origin of cfDNA in sepsis.** An increase in total cfDNA levels in septic patients has been previously documented, and even shown to have a prognostic value[33,34]. However, it is unclear which cell types are contributing to the elevated cfDNA. We analyzed the cfDNA methylation profile of 14 samples from patients with sepsis. In most patients (13/14) the main contributors to the increase in cfDNA were leukocytes (mainly granulocytes), elevated >20-fold relative to healthy levels (Fig. 6a, b). In some cases, varying amounts of hepatocyte cfDNA were detected (patients SEP-026, SEP-017, SEP-016). Importantly, the levels of hepatocyte cfDNA were strongly correlated (Pearson's

$r = 0.931$, $p < 5e−7$) with levels of alanine aminotransferase (ALT) in circulation, a marker of hepatocyte damage (Fig. 6c).

**Identifying tumor origin by cfDNA methylation.** We deconvoluted the cfDNA methylation profiles of patients with metastatic colon cancer ($n = 4$), lung cancer ($n = 4$), and breast cancer ($n = 3$) (Supplementary Data 1). All had elevated concentration of cfDNA compared to healthy individuals (>20-fold increase). The tissue of origin was the strongest signal (most genome equivalents/ml) in the majority of cases (8/11 total, 3/4 colon, 2/4 lung, 3/3 breast, Fig. 7a–c). These findings indicate the ability of the deconvolution algorithm to correctly detect cfDNA from advanced cancer, despite potential changes to the epigenome of cancer cells.

To assess the accuracy of cancer detection using deconvolution, we performed a mixing experiment, where plasma from a patient with colon cancer was mixed with plasma of healthy donors at different proportions (Supplementary Data 1), and the methylome of the resulting mixture was deconvoluted. The algorithm correctly identified the presence of colon DNA in the mixes, in the correct proportion, down to 3% (33-fold dilution of the original cancer plasma sample, $r = 0.92$, $p < 1.2e−3$) (Fig. 7d).

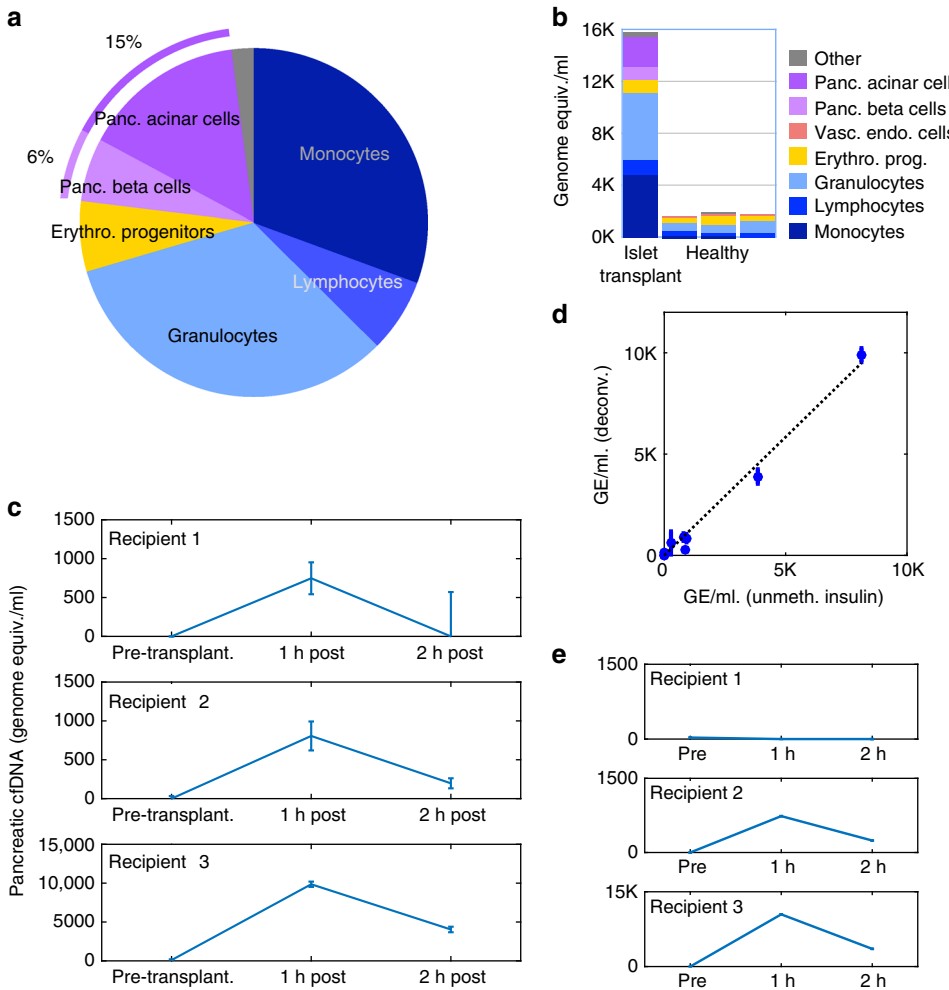

**Fig. 5** Cellular contributors to cfDNA in islet transplant recipients. **a** Deconvolution results for pooled sample of cfDNA from five patients, 1 h after islet transplantation. The patients present a noticeable amount of pancreas-derived cfDNA (typically absent in healthy donors). Cell types contributing <1% were included in "Other". **b** Same as **a**, expressed as absolute levels of cfDNA (genome equivalents per ml plasma). Also shown is the prediction for a healthy individual. **c** Inferred amount of cfDNA from all three pancreas cell types for three individuals prior to, 1 h after, and 2 h after islet transplantation. Error bars: SD, estimated using Bootstrapping. **d** Comparison of pancreatic cfDNA estimations using deconvolution (y-axis) to results of targeted insulin promoter methylation assay (x-axis). Pearson's $r = 0.996$, $p$-value $= 1.6e{-}8$. **e** Same as **c**, using a reference atlas with whole pancreas methylome, instead of purified pancreas cell types. Here, deconvolution fails to identify pancreatic cfDNA in recipient 1

To further assess the performance of the deconvolution algorithm, we applied it to recently published dataset where plasma samples of prostate cancer patients were assessed using Illumina 450K arrays, before and after treatment with Abiraterone Acetate, including patients that were responsive or not responsive to therapy[35]. As shown in Fig. 7e, the algorithm detected prostate DNA in most patients (as compared to a lack of signal in all healthy controls). Strikingly, the deconvolution algorithm also detected a sharp decline in the levels of prostate cfDNA in treatment-sensitive patients ($p < 0.019$, paired $t$-test) but not in treatment-resistant patients ($p < 0.909$, paired $t$-test), further supporting validity of the method.

Finally, we tested whether an unbiased deconvolution approach could be useful in identifying a cancer tissue of origin, even in the absence of an identifiable primary tumor. To this end, we analyzed the plasma cfDNA of four patients with Cancer of Unknown Primary (CUP). All patients had metastatic disease with no clear pathological identification of the primary source of cancer (detailed in Supplementary Data 1). In each case the

suspected origin of the tumor, based on clinical history and pathology reports, showed a strong signal in the deconvolution analysis (Fig. 7f). Patient 3, for example, presented with metastases in bones and lungs without identifiable histopathology. Six years earlier, the patient had a local bladder carcinoma that was treated and removed. Deconvolution analysis of plasma cfDNA identified a significant contribution by bladder cells (>5000 genome equiv./ml), suggesting that the current disease originated from previously disseminated bladder cancer cells (Fig. 7f).

These findings indicate that cfDNA methylation deconvolution can be the basis of a non-invasive approach to identify the origin of cancer, similar to what has been described using biopsy material[36].

## Discussion

In many diseases, DNA from dying cells is released into the bloodstream. Tools that can identify the source tissue of this

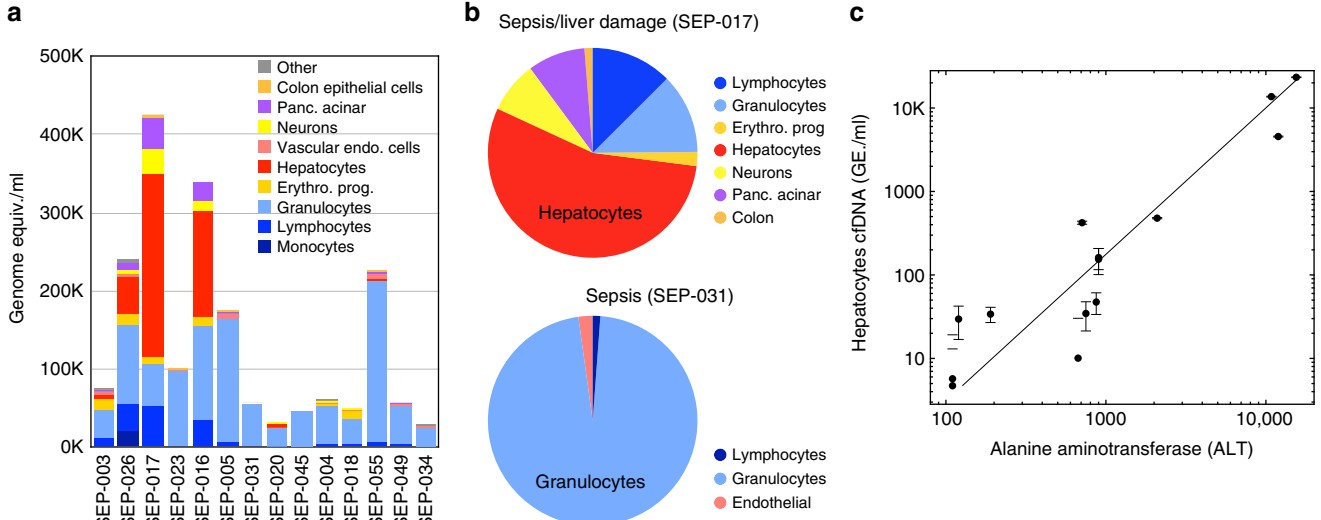

**Fig. 6** Cellular contributors to cfDNA in sepsis. **a** Predicted cellular contributions are shown for 14 samples of cfDNA from patients with sepsis. Cell types present at <1% were included in "Other". **b** Pie charts representing predicted distribution of cell types contributing to cfDNA of two of the sepsis patients. **c** Predicted levels of hepatocyte cfDNA compared to serum levels of alanine aminotransferase (ALT), a standard biomarker for hepatocyte damage (Pearsons's $r = 0.93$, $p$-value $\leq 4e{-}7$). Error bars: SD, as estimated using Bootstrapping

DNA could be instrumental in identifying and locating disease. DNA methylation reflects cell identity, and is therefore an ideal marker of the origin of DNA in circulation. In this study, we present a method to decipher the cellular origins of cfDNA by deconvoluting genome-wide methylation profiles, and use it to determine which cells release DNA into blood in several clinically relevant situations.

When assessing the tissues that contribute to human cfDNA, we first made an effort to define the healthy baseline. Previous studies used plasma from female patients who had received bone marrow transplants from male donors, and concluded that most cfDNA is derived from cells of hematopoietic origin[32]; however, the contribution of individual blood cell types was not assessed, nor was the contribution of non-blood cells. More recently, Guo et al. analyzed the plasma methylome of healthy and cancer patients using WGBS, and reported the contribution of white blood cells (without subtypes) as well as nine solid tissues and two tumor types[20]. Our deconvolution assay revealed the specific contributors to healthy plasma, namely granulocytes, monocytes, lymphocytes, and erythrocyte progenitors. The latter is consistent with a recent report that used specific erythroid lineage methylation markers to identify erythroid lineage-derived cfDNA[15]. Note that unlike the other sources of cfDNA, in this case the process reflected by cfDNA might be cell birth (the generation of enucleated red blood cells) rather than cell death. Refinement of the methylome atlas will likely result in further refinement of cfDNA interpretation, even retrospectively on the samples reported here. For example, it should be possible to determine the relative contribution of neutrophils and other cell types to the granulocyte cfDNA pool, and of circulating monocytes and tissue resident macrophages to the monocyte cfDNA pool.

Beyond blood cells, we found that ~10% of cfDNA in healthy individuals is derived from vascular endothelial cells (a finding made possible by the generation of a vascular endothelial cell methylome reference), and that ~1% of cfDNA is derived from hepatocytes, which is consistent with our recent observation of hepatocyte cfDNA in healthy plasma using three targeted hepatocyte markers[18]. The cfDNA signal from the vasculature and the liver reflects the sum of multiple parameters: total cell number in

these organs, the degree of baseline turnover, and the fact that cfDNA from these tissues is apparently cleared via blood. The absence of a cfDNA signal from other tissues in the body, known to have a high turnover rate, likely reflects alternative clearance routes: for example, dying intestinal epithelial cells under healthy conditions likely shed cfDNA into the lumen of the intestine, rather than to blood. Similar considerations apply to the lung, kidney and skin. The algorithm also detected a neuronal-derived signal comprising as much as ~2% of the healthy plasma methylome. While this finding may reflect a baseline turnover of central or peripheral neurons[37], we cannot rule out the possibility that it is an artifact of the deconvolution algorithm, due to a partial and imperfect reference atlas. One argument in favor of the latter interpretation is that our directed PCR-sequencing assays using brain-specific methylation markers show only a negligible neuronal signal in healthy individuals (~0.1%), while positive controls with brain damage do show a clear signal (manuscript in preparation and ref. [11]). More experiments are needed to determine the actual contribution of neuronal DNA to the healthy cfDNA.

We also performed a preliminary analysis of cfDNA composition as a function of age, using pools of samples from healthy individuals aged 75 and above and between the ages of 19 and 30. Two striking findings emerge from the analysis of these samples: first, the total concentration of cfDNA in aged individuals is about twice that of people in their 3rd decade of life; second, deconvolution revealed a distribution of sources that is highly similar between aged and young individuals. We propose that this similarity reflects a decrease in the rate of cfDNA clearance in old age, rather than a concordant increase in cell death within all tissues. Additional studies are required to definitively interpret the biology of the circulating methylome in old age.

The application of cfDNA deconvolution to selected pathologies provided further support as to the validity of the approach. This included the identification of pancreas cfDNA in islet transplant recipients (but not in healthy controls) and the identification of elevated hepatocyte cfDNA in patients with sepsis, which correlated with an independent circulating liver marker. In both transplantation and sepsis we found that elevated cfDNA

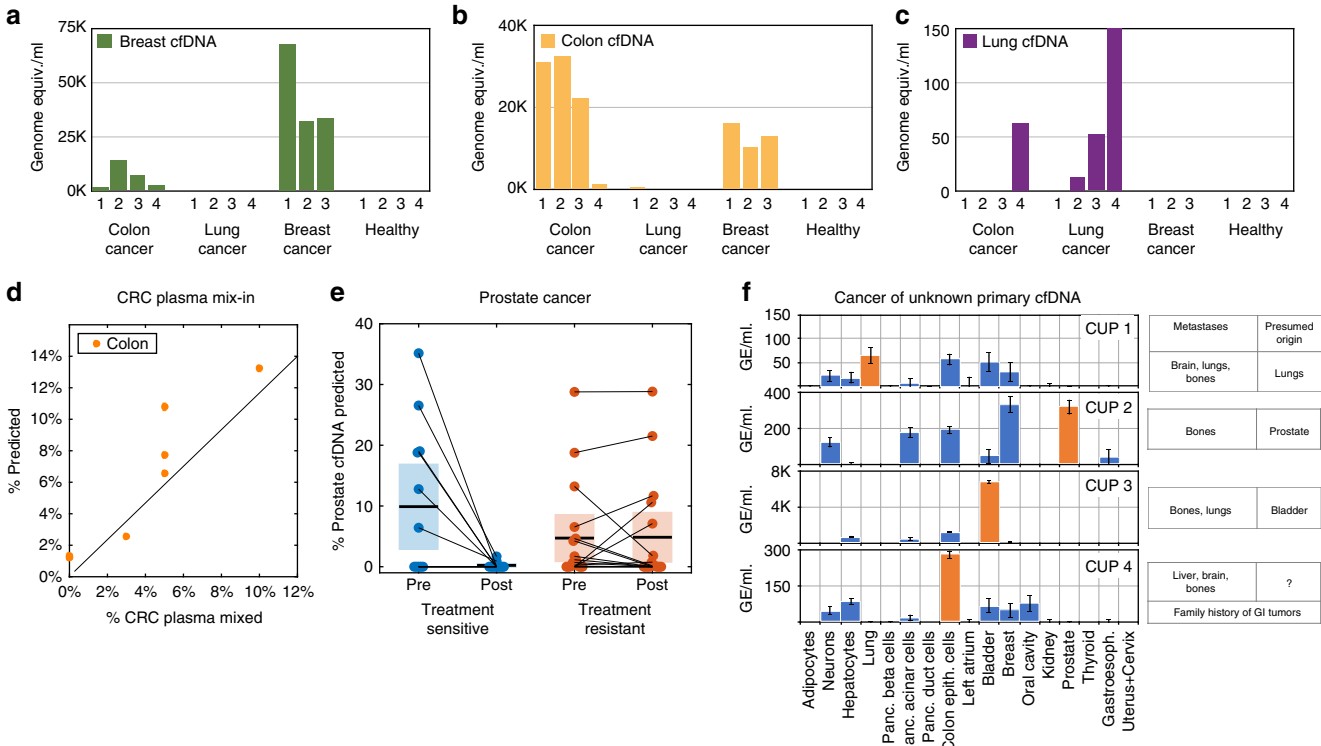

**Fig. 7** Cellular contributors to cfDNA in cancer. **a–c** Predicted contributions of breast, colon, and lung DNA to the plasma methylome of four patients with colon cancer (CC), four patients with lung cancer (LC), three patients with breast cancer (BRC), and four healthy donors (H). All patients were at advanced stages of disease. **d** A mix-in experiment. The plasma of a patient with advanced colon cancer was mixed with three healthy plasma samples in varying proportions (detailed in Supplementary Data 1), and the fraction of colon-derived cfDNA was assessed using deconvolution of the methylome. **e** Identification of prostate-derived cfDNA in published plasma methylomes of patients with prostate cancer[35] before and after treatment. Patients classified as abiraterone acetate (AA) treatment responsive (blue) show a dramatic drop in prostate-derived cfDNA, compared with the AA-resistant patients (red). **f** Deconvolution of cfDNA methylation predicts cfDNA origin for CUP cancer patients. Shown are the predicted cellular contributors for cfDNA samples from four patients diagnosed with a Cancer of Unknown Primary (CUP). Blood cell types and cells contributing <1% are not shown. For each patient, the location of metastases and the presumed tissue source of cancer according to clinical history are listed. Deconvolution results agreeing with clinical predictions are shown as orange bars. Error bars: SD, as estimated using Bootstrapping

was mostly derived from immune cells. Both scenarios likely involve strong immune reactions and the increase in leukocyte-derived cfDNA may be derived from cells that died during cell division or as part of an immune response. We also demonstrated that deconvolution can identify cfDNA from a cancer's tissue of origin, even in advanced tumors presumably presenting with epigenomic instability. While more studies with plasma samples from cancer patients are needed, in particular from early stage diseases, our findings from multiple type of cancer (colon, lung, breast, and prostate) are highly encouraging in this respect. Lastly, using plasma samples from patients with cancer of unknown primary, we showed that the tissue source of metastases can be identified by analysis of cfDNA methylation, even in cases where the primary tissue of the cancer is missing and unclear. Whilst most current approaches aim to monitor cancer via identification of mutations in cfDNA, we propose that combining such an analysis with cfDNA methylation deconvolution may eventually allow for early and unbiased diagnosis of cancer and its location[7].

Our work provides a proof of concept for the utility of plasma methylome deconvolution in studying human tissue dynamics in health and disease, adding insights beyond those of recent reports in this emerging field[19–22]. Furthermore, our approach can easily be adapted to determine the cellular contributors to cfDNA in virtually any setting in which there is a question regarding the

composition of cfDNA. We selected to work with Illumina arrays as a platform for both the tissue reference atlas and the plasma methylome assay. This platform has multiple advantages, perhaps most importantly the vast amount of public data available that can be used to construct a tissue methylome atlas. Additionally, it is the most affordable method available for obtaining high-resolution genome-wide methylation profiles and is simple to perform and analyze as well as scalable. However, arrays have also important limitations: they cover only a small fraction of the genome-wide methylome; they report on the methylation status of individual CpG sites, missing the information embedded in the status of methylation haplotype blocks[11,20]; they suffer from batch effects; they require a relatively large amount of DNA (100 ng cfDNA, shown here to be sufficient for deconvolution, requires about 40 ml of blood); and their sensitivity (ability to detect a small fraction of molecules with a different methylation status in a mixture) is limited compared with sequencing of individual molecules. We believe that in the long run, for applications requiring maximal sensitivity and affordability (such as for early detection of cancer in asymptomatic individuals), a cfDNA methylation deconvolution approach based on deep sequencing of a collection of informative CpG blocks, possibly following capture of key loci from plasma, using a sequencing-based comprehensive atlas, will likely be the preferred approach.

Nonetheless, our study does provide some important insights into design principles of effective plasma methylome technology, which are general and would hold for other platforms including massively parallel bisulfite sequencing or nanopore sequencing. These include: (1) the key importance of generating a comprehensive methylation atlas composed of individual cell types (purified from fresh tissue), rather than whole tissues. The inclusion of cell-type specific methylomes allows the identification of important tissue contributions to cfDNA, including cell types that comprise a small minority of their host tissue (e.g. beta cells in the pancreas), and cell types that are present within multiple organs and hence might be masked (e.g. vascular endothelial cells). (2) Not all CpG sites contribute to accurate deconvolution; in fact, deconvolution based on a defined subset of informative sites performs better than an approach taking into account all sites, including those that are not differentially methylated between tissues and hence contribute mostly noise; (3) a specific subset of ~4000 CpG sites that is informative enough for accurate estimation of cfDNA contributors. We propose that a capture-based approach, applying deep bisulfite sequencing to probe multiple neighboring CpGs from the same molecule around selected loci, would offer deconvolution at a much greater resolution, and potentially using lower amount of DNA.

In summary, we report a method for interpreting the circulating methylome using a reference methylome atlas, allowing inference of tissue origins of cfDNA in a specific and sensitive manner. We propose that deconvolution of the plasma methylome is a powerful tool for studying healthy human tissue dynamics and for identifying and monitoring a wide range of pathologies.

## Methods

**Reference matrix**. All DNA methylation profiles were determined either on the Illumina Infinium Human Methylation 450K or EPIC BeadChip arrays. DNA methylation data for white blood cells (neutrophils, monocytes, B-cells, CD4+ T-cells, CD8+ T-cells, NK-cells, $n = 6$ each) were downloaded from GSE110555 (EPIC)[38]. Data for erythrocyte progenitors ($n = 5$) were downloaded from GSE63409 (450K)[39], and data for left atrium ($n = 4$) were downloaded from GSE62727 (450K)[40]. Data for bladder ($n = 19$), breast ($n = 98$), cervix ($n = 3$), colon ($n = 38$), esophagus ($n = 16$), oral cavity ($n = 34$), kidney ($n = 160$), prostate ($n = 50$), rectum ($n = 7$), stomach ($n = 2$), thyroid ($n = 56$), and uterus ($n = 34$) were downloaded from TCGA[26]. DNA methylation data for adipocytes ($n = 3$, 450K), hepatocytes ($n = 3$, 450K and EPIC), alveolar lung cells ($n = 3$, EPIC), neurons ($n = 3$, 450K and EPIC), vascular endothelial cells ($n = 2$, EPIC) pancreatic acinar cells ($n = 3$, 450K and EPIC), duct cells ($n = 3$, 450K and EPIC), beta cells ($n = 4$, 450K and EPIC), colon epithelial cells ($n = 3$, EPIC) were generated in house and are available from the corresponding authors upon reasonable request. Detailed sample information is available in Supplementary Data 1.

**Cell isolation**. Cancer-free primary human tissue was obtained from consenting donors, dissociated to single cells, sorted using cell type-specific antibodies and lysed to obtain genomic DNA, from which 250 ng were applied to an Illumina methylation array. Adipocytes ($n = 3$) were isolated from fat tissue according to the collagenase procedure of Rodbell[41]. In brief, tissue was cut into ≈20 mg pieces and incubated (10 g tissue/25 ml buffer) in Krebs-Ringer phosphate (KRP buffer, pH 7.4) containing 4% bovine serum albumin (BSA) and 0.5 mg/ml of collagenase type 1 for 45 min at 37 °C in a shaking water bath. The isolated adipocytes were collected through a 250 μm nylon mesh filter and were washed 3-4 times with 1% KRP-BSA washing buffer. The stromal vascular fraction (SVF) in the washing buffer was collected by $500 \times g$ centrifuge at 4 °C for 10 min. Cells were then homogenized in lysis buffer (0.32 M sucrose, 25 mM KCl, 5 mM MgCl2, 0.1 mM EDTA, 10 mM Tris-HCl pH 7.5, 0.005% NP-40, 1 mM DTT) transferred to ultracentrifuge tubes, layered onto a sucrose cushion solution (1.8 M sucrose, 25 mM KCl, 5 mM MgCl2, 0.1 mM EDTA, 10 mM Tris-HCl pH 7.5, 1 mM DTT) and centrifuged at $106,750 \times g$ for 1 h at 4 °C to isolate nuclei. Cortical neurons ($n = 1$) were isolated from human occipital cortex by sucrose-gradient centrifugation and labeled with Alexa Fluor 647 conjugate of neuron-specific monoclonal anti-NeuN antibody (A-60) (Millipore, 1:1000). NeuN-positive and negative nuclei were sorted by FACS and DNA was extracted[42,43]. Hepatocytes ($n = 2$) were isolated as previously described[44]. Pancreatic acinar cells and duct cells ($n = 3$) were obtained from cadaveric donors as described[45]. Pancreatic beta cells ($n = 4$) were isolated from cadaveric islets as previously described[46]. Vascular endothelial cells were

isolated from the saphenous vein, surgically excised due to chronic insufficiency. Dissociated endothelial cells were captured using mouse anti-human CD105 magnetic beads (cat #130-051-201, Miltenyi, 1:5) ($n = 3$ donors, pooled to 2 samples, one containing material from two donors and one containing material from one sample). Distal lung tissue ($n = 3$ donors, 3 samples) was dissociated using an adaptation of previous protocols[47–50]. Briefly, alveolar epithelial cells were enriched using mouse anti-human CD105 magnetic beads for depletion of endothelial cells (cat #130-051-201, Miltenyi, 1:5) and subsequently mouse anti-human Epcam (CD326) magnetic beads to capture epithelial cells (cat #130-061-101, Miltenyi, 1:4) or by FACS sorting using the following antibodies: CD45 eFluor 450 (cat #48-9459-41), CD31 eFluor 450 (cat #48-0319-42) and CD235a eFluor 450 (cat #48-9987-42) (all from eBioscience, 1:20) and CD326-APC (cat #130-113-260, Miltenyi, 1:50). Colon epithelial cells were dissociated using an adaptation of a published protocol[51] and were sorted by FACS using CD45 eFluor 450 (cat #48-9459-41), CD31 eFluor 450 (cat #48-0319-42) and CD235a eFluor 450 (cat #48-9987-42, eBioscience, 1:20) (for blood and endothelial cell lineage depletion), and CD326-APC (Miltenyi, 1:50, cat #130-113-260) antibodies. FACS gating strategies are shown in Supplementary Figure 8.

**Blood samples**. Donors were consented and whole blood (usually 20 ml) was drawn, collected into an EDTA tube, and spun quickly to separate plasma, which was stored at −20 °C until isolation of cfDNA.

**Human research participants**. Tissue and plasma samples were obtained in accordance with the principles endorsed by the Declaration of Helsinki and written informed consent was obtained from all subjects. Protocols were approved by the Institutional review boards of Hadassah-Hebrew University Medical Center, The University of Alberta, Karolinska Institute and Oregon Health & Science University.

**Sample pooling**. Pooled DNA samples were obtained by mixing DNA from several individuals. DNA was extracted from 8ml of plasma and samples were added until 250 ng reached (7–19 samples per pool). No individual contributed more than two times as much DNA to a pool than another individual.

**DNA extraction**. 250 ng was collected from each sample, except where otherwise specified. DNA concentration was measured with Qubit. cfDNA extraction from plasma was performed with the QIAsymphony liquid handling robot. cfDNA was treated with the Illumina Infinium FFPE restoration kit and hybridized to the Illumina 450K or EPIC arrays.

For adipocytes, we used a modified protocol from Miller et al.[52]. Five hundred microliters DNA lysis buffer (200 mM NaCl, 5 mM EDTA, 100 mM Tris-HCl pH 8, 1 % SDS) and 6 μl Proteinase K (20 mg/ml) were added to the collected nuclei and incubated at 55 °C overnight. RNase cocktail (Ambion) was then added and incubated at 55 °C for 1 h. Half of the existing volume of 5 M NaCl solution was added and the mixture agitated for 15 s. The solution was spun down at $16,000 \times g$ for 3 min. The supernatant containing DNA was transferred to a new Eppendorf tube. Three times of the existing volume of 95% ethanol was added and the tube was inverted several times to precipitate adipocytes or SVF DNA. The DNA precipitate was washed three times in 75% ethanol and air-dried at 55 °C for 2 h. 500 μl DNase/RNase-free water was used to suspended the dried DNA. All DNA samples were quantified and purity-checked by UV spectroscopy (Nanodrop).

Neuronal DNA was extracted by adding 500 μl DNA lysis buffer (100 mM Tris-HCl [pH 8.0], 200 mM NaCl, 1% SDS, and 5 mM EDTA) and 6 μl Proteinase K (20 mg/ml, Invitrogen) to the sorted nuclei and incubated overnight at 65 °C. Following overnight incubation, an RNase cocktail was added (3 μl, Ambion) and incubated at 65 °C for 45 min. Half of the existing volume of 5 M NaCl solution was added and the mixture agitated for 15 s and centrifuged at $16,000 \times g$ for 3 min. The supernatant containing the DNA was transferred to a 12 ml glass vial. Three times the volume of 95% ethanol was added to the glass vial and inverted several times to precipitate the DNA. The DNA precipitate was washed in DNA-washing solution (70% [v/v] ethanol and 0.5 M NaCl) for 15 min for three times and transferred to 200 μl DNase-/ RNase-free water (Gibco/Life Technologies) and air-dried at 65 °C overnight. Finally, the DNA was dissolved in 500 μl TE buffer (pH 8.0) (10 mM Tris-HCl [pH 8.0] and 1 mM EDTA). The DNA was quantified and its purity was verified using a NanoDrop 2000 spectrophotometer (ThermoScientific).

**Data processing**. Methylation array data were processed with the minfi package in R. For each sample analyzed on the Illumina Methylation array, CpG sites were filtered out if they were represented by less than 3 beads on the array, if the detection $p$-value (representing total fluorescence of the relevant probes) was >0.01, or if they mapped to a sex chromosome. Background correction and normalization were performed with the preprocessIllumina function, which removes background calculated based on internal control probes and normalizes all samples to a pre-determined control sample.

 

**Comparison of EPIC and 450K platforms.** As the reference database included samples analyzed with two highly similar yet not identical platforms, the Illumina 450K array and the Illumina EPIC array, we looked to identify and remove sites with low reproducibility between the platforms. To this end, we collected data from samples analyzed on both platforms: 15 samples from GSE86833[53], 12 samples from GSE92580[54], and one sample from our generated dataset (hepatocytes). For each overlapping CpG, we then calculated the median absolute error (MAE) between the 450K samples and the corresponding EPIC samples, and removed 37,747 CpGs with an MAE > 0.05.

**CpG feature selection.** First, CpGs whose variance across the entire methylation atlas was below 0.1%, or CpGs with missing values were excluded. We then selected the $K = 100$ most specific hypermethylated CpGs for each cell type. Let us denote the methylation matrix $\mathbf{X}$, composed of $N$ rows (CpGs) by $d$ columns (cell types). We then divided each row (the methylation pattern of one CpG over all cell types) by its sum $\mathbf{X}_i' = \frac{\mathbf{X}_i}{\sum_j^d \mathbf{X}_{ij}}$. For each cell type $j$, we identified the top $K$ hyper-methylated CpGs with the highest $\mathbf{X}'_{i,j}$ values. To identify uniquely hypomethylated CpGs, we performed a similar process for the reversed methylation matrix $(\mathbf{1} - \mathbf{X})$. Finally, for each cell type we included both the top $K$ hypermethylated and the top $K$ unmethylated CpGs in the reference matrix (Supplementary Data 1). To this set of CpGs, we added neighboring CpGs, up to 50 bp.

Pairwise-specific CpGs were iteratively selected as follows: given the current set S of CpGs, we projected the reference atlas on those coordinates, and calculated the Euclidean distances between pairs of cell types. Once the closest pair of cell types was identified, we selected the CpG site where they differ the most, and added it into the set S. This process was iteratively repeated, focusing on the most confusing pair of cell types in each iteration.

**Deconvolution.** To calculate the relative contribution of each cell type to a given sample, we performed non-negative least squares, as implemented in the nnls package in MATLAB (an efficient alternative to lsqnonneg). Given a matrix $\mathbf{X}$ of reference methylation values with $N$ CpGs and $d$ cell types, and a vector $\mathbf{Y}$ of methylation values of length $N$, we identified non-negative coefficients $\beta$, by solving $\text{argmin}_\beta \|\mathbf{X}\beta - \mathbf{Y}\|_2$, subject to $\beta \geq 0$. We then adjusted the resulting $\beta$ to have a sum of 1, where for each $\beta_j$ we defined $\beta_j' = \frac{\beta_j}{\sum_j^d \beta_j}$. To obtain absolute levels of cfDNA (genome equivalent/ml) per cell type, we multiplied the resulting $\beta_j'$ by the total concentration of cfDNA present in the sample, as measured by Qubit. It was assumed that the mass of a haploid genome is 3.3 pg and as such, the concentration of cfDNA could be converted from units of ng/ml to haploid genome equivalents/ml by multiplying by a factor of 303. To estimate deconvolution error rates, we used a bootstrap approach, where we also analyzed the observation vector ($\mathbf{Y}$) using $n = 100$ instances of the methylation atlas. Following Houseman et al.[30], and due to the limited number of replicate per cell type, we used a parametric approach, where the original replicates for each tissue were used to estimate the mean CpG methylation and its standard deviation. We then generated $n = 100$ new methylation atlases ($\mathbf{X}'$) by sampling from Normal distributions centered at these values for each CpG/tissue. Finally, we deconvoluted the observation vector ($\mathbf{Y}$) using each atlas, and estimated the empirical standard deviation of the admixture parameters across atlases ($\mathbf{X}'$). The same approach was used to estimate the variation for contribution of specific cell types, including DNA mixes (Fig. 3a–d), pancreas (Fig. 5c–e), hepatocytes (Fig. 6c), and plasma mixes (Fig. 7d).

**Simulations.** We analyzed 18 leukocyte samples (whole-blood) with Illumina methylation EPIC arrays. For each cell type, we mixed in every analyzed replicate with each leukocyte sample in ratios of 0 to 100, 0.1 to 99.9, 1 to 99, 2 to 98, etc. up to 10 to 90. For every combination of leukocytes and cell type replicate, we updated the reference atlas by excluding the mixed-in sample and then re-computing the average methylome for that cell type using all other replicates. We then re-applied the feature selection process (using the new atlas), applied the deconvolution algorithm, and estimated the admixture coefficients for all cell types. This procedure ensures that the training set is completely separated from the test set. Finally, we calculated for each cell type, at each admixture ratio, the average predicted proportion over all replicates, its median, and the range between the 1st and 3rd quartiles.

**Reproducibility.** We assayed three cfDNA samples in duplicate (Supplementary Figure 7a-c). The predicted proportions of cell types contributing to the samples were highly correlated ($r > 0.99$). Furthermore, as the amount of cfDNA available is often limited, we also evaluated the possibility of using less than the 250 ng cfDNA (as recommended by Illumina for analysis with methylation array). The results were reproducible with as little as 50 ng of cfDNA ($r > 0.9$) (Supplementary Figure 7a-d).

**Code availability.** A standalone program for deconvolution of array methylome is available at https://github.com/nloyfer/meth_atlas or from the corresponding authors.

## Data availability

The datasets generated and analyzed during this study are summarized in Supplementary Data 1 and available at NCBI Gene Expression Omnibus (GEO) database repository with the dataset identifier GSE122126. A Reporting Summary for this Article is available as a Supplementary Information file.

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

## Acknowledgements

This research was performed using grants from The Ernest and Bonnie Beutler Research Program of Excellence in Genomic Medicine, the Juvenile Diabetes Research Foundation, The Alex U Soyka pancreatic cancer fund (to Y.D.), the NIDDK-supported Human Islet Research Network (HIRN) (RRID:SCR_014393; https://hirnetwork.org; UC4 DK104216-01), the Kahn Foundation (to B.G., R.S., and Y.D.) and GRAIL (to B.G., R.S., T.K., and Y. D.). T.K. was supported by the Israeli Centers of Excellence (I-CORE) for Gene Regulation in Complex Human Disease (no. 41/11) and Chromatin and RNA in Gene Regulation (no. 1796/12), and by Israel Science Foundation grant (no. 913/15). N.L. was supported by a fellowship from the Leibnitz Center for Research in Computer Science. K. L.S. and P.A. acknowledge the Swedish Research Foundation, Novo Nordisk Foundation and the Diabetes Research Program at Karolinska Institutet. K.L.S. also acknowledges the KI-AZ ICMC and the Vallee Foundation. Illumina array experiments were performed with the help of the Roswell Park Cancer Institute.

## Author contributions

Conceived and designed the methods: J. Moss, B.G., R.S., T.K. and Y.D.; data collection and contribution: J. Moss, J. Magenheim, D.N., H.Z., A.K., Y.S., M.M., H.D., P.A., K.-Y.F., E.K., K.L.S., G.L., A.Z., A.G., A.M.J.S., M.G., A.D.W., B.G. and R.S.; analyzed the data: J. Moss, N.L. and T.K.; wrote the paper: J. Moss, B.G., R.S., T.K. and Y.D.

## Additional information

**Competing interests:** J.M., R.S., B.G., T.K. and Y.D. are inventors on a patent entitled "CELL FREE DNA DECONVOLUTION AND USE THEREOF" (US provisional application No. 62/661,179). The remaining authors declare no competing interests.

