## [Peer Review File · Nature Communications]

Reviewer #1 (Remarks to the Author):

The manuscript presents important novel results about the contribution of different cell-types to cfDNA in health and pathology, as well as a roadmap to future work in this area that will expand the atlas of methylomes for individual cell-types that can make this approach more powerful and accurate.

The authors have addressed key points and I support publication.

Authors should consider below comments.

In the introduction, would be suitable to refer to work by Ulz et al., Inferring expressed genes by whole-genome sequencing of plasma DNA, Nature Genetics 2016. The contribution of proliferative processes to the origin of cfDNA can be reflected more clearly in the introductory remarks.

The simulated mixtures, as presented in Figure 2, contain many assumptions hidden in the way these were generated. They show presence of cell-type markers, already shown in figure 1, and that the deconvolution process works. But don't support conclusions about sensitivity in a real biological sample with DNA from different origins subject to measurement limitations and errors.

Mixtures of cell-lines in Figure 3 support the ability to detect identifiable markers. But the processing of these samples is very different from blood samples and this can't be used to infer sensitivity. The following figures are more convincing.

Figure 4, add legend for the yellow section in part a, add data for Other in part b to clarify the total which is higher for >75 yrs. Replace young/old by more explicit labels, 19-30 yr and >75 yr. Text says >67 yr.

Are the two datasets for each case technical duplicates or different sample pools? How many samples/individuals in each pool?

Line 193-194 claim a signal for neurons. The deconvolution is unable to identify cell types other than the 25 and it is possible that this signal is from something else not yet learned. Data alluded to in

lines 313-316 makes this possibility stronger. The statement in 193-194 needs to reflect that, stating that this a signal was identified as neurons by the deconvolution but may be or is likely to be of different origin.

Figure 5, please add also time-course data for the cfDNA of immune cell origin.

Figure 6a: 'other' not shown. 6c: error bars by bootstrapping, not clear how the bootstrapping is done.

The point in lines 320-322, 'first, the total concentration of cfDNA in aged individuals is about twice that of people in their 3rd decade of life' is also of value in other areas. The way this is presented only after the deconvolution and then adding up the contribution makes this more difficult to compare to other sources of data. Authors should consider adding a supplementary figure showing only total levels in these individuals, explaining how this data was generated using a simpler method or metric.

Reviewer #2 (Remarks to the Author):

The authors have addressed all my previous comments. I have only one comment about data presentation: Figure 3, 4d and 7d (as well as similar plots) need a diagonal reference line just like in Figure 2a such that the goodness of fit is more evident.

Reviewer #3 (Remarks to the Author):

The authors have made some important improvements to the manuscript, following the previous round of review.

I still have some concerns:

- Each mixture for a given cell type of the in-vitro mixing experiments (Fig 3) should consist of cells from a different set of individuals in order to provide a realistic test of the method. It was not clear whether this was the case. If the same cells are being mixed in different proportions then the experiment lacks the inter-individual variation that makes real applications of deconvolution harder. From the legend, at least the blood cells are from a single individual, and the target cell types may also be.

- The new Figure 3 should include the $x=y$ line.

- I don't feel the authors responded to the point in my original review regarding the bootstrapping method that was used to produce error bars of Figs 5c-e and Fig. 6c. The bootstrap procedure described will have the effect of assessing the robustness of their result to the choice of CpGs.

- The issue concerning bootstrapping on CpGs also appears to arise in the new figures (3 and 7d)

Detailed responses to reviewer comments

Reviewer #1:

The manuscript presents important novel results about the contribution of different cell-types to cfDNA in health and pathology, as well as a roadmap to future work in this area that will expand the atlas of methylomes for individual cell-types that can make this approach more powerful and accurate.

The authors have addressed key points and I support publication.

We thank the reviewer for this positive evaluation.

In the introduction, would be suitable to refer to work by Ulz et al., Inferring expressed genes by whole-genome sequencing of plasma DNA, Nature Genetics 2016.

We have now included a reference to Ulz et al to the Introduction.

The contribution of proliferative processes to the origin of cfDNA can be reflected more clearly in the introductory remarks.

Done.

The simulated mixtures, as presented in Figure 2, contain many assumptions hidden in the way these were generated. They show presence of cell-type markers, already shown in figure 1, and that the deconvolution process works. But don't support conclusions about sensitivity in a real biological sample with DNA from different origins subject to measurement limitations and errors.

Mixtures of cell-lines in Figure 3 support the ability to detect identifiable markers. But the processing of these samples is very different from blood samples and this can't be used to infer sensitivity. The following figures are more convincing.

Thank you. We agree with this assessment. The *in silico* simulations and the *in vitro* mixes were necessary steps laying the ground for the *in vivo* experiments, which indeed reflect the real biological substrate of cfDNA but have inherent limitations, e.g. the quantitative contribution of each cell type is not known.

Figure 4, add legend for the yellow section in part a, add data for Other in part b to clarify the total which is higher for >75 yrs. Replace young/old by more explicit labels, 19-30 yr and >75 yr. Text says >67 yr.

Are the two datasets for each case technical duplicates or different sample pools? How many samples/individuals in each pool?

Thank you. We have now added a legend for the yellow section in Fig4a (neurons); the “Other” section in Fig4b; and replaced the age labels as requested.

The two datasets are biological replicates: each is a pool of different individuals.

We have now added a new Supplemental Table (new Table 3 of Supp File 1), listing the ages of all donors in each pool. The average ages of the “old” samples are greater than 75 but the youngest individual is 67, which was the source of the confusion. All the young donors were between 19-30 years of age.

Line 193-194 claim a signal for neurons. The deconvolution is unable to identify cell types other than the 25 and it is possible that this signal is from something else not yet learned. Data alluded to in lines 313-316 makes this possibility stronger. The statement in 193-194 needs to reflect that, stating that this a signal was identified as neurons by the deconvolution but may be or is likely to be of different origin.

Thank you. We have revised the text per the reviewer’s request.

Figure 5, please add also time-course data for the cfDNA of immune cell origin.

We have now added a new supplemental figure (S6), showing the inferred distinct dynamics of immune cell cfDNA in the plasma of pancreatic transplantation patients.

Figure 6a: ‘other’ not shown. 6c: error bars by bootstrapping, not clear how the bootstrapping is done.

We have now added an “Other” category to Figure 6a, and elaborated on the bootstrapping procedure from which we computed the error bars.

The point in lines 320-322, ‘first, the total concentration of cfDNA in aged individuals is about twice that of people in their 3rd decade of life’ is also of value in other areas. The way this is presented only after the deconvolution and then adding up the contribution makes this more difficult to compare to other sources of data. Authors should consider adding a supplementary figure showing only total levels in these individuals, explaining how this data was generated using a simpler method or metric.

We thank the reviewer for this suggestion, and have now added a new Supplemental Figure S5, showing the total levels of cfDNA (Genome equivalents/ml plasma) in these individuals, and referred to it in main text.

Reviewer #2:

The authors have addressed all my previous comments. I have only one comment about data presentation: Figure 3, 4d and 7d (as well as similar plots) need a diagonal reference line just like in Figure 2a such that the goodness of fit is more evident.

We thank the reviewer, and have now added the requested reference lines.

Reviewer #3

The authors have made some important improvements to the manuscript, following the previous round of review. I still have some concerns:

- Each mixture for a given cell type of the in-vitro mixing experiments (Fig 3) should consist of cells from a different set of individuals in order to provide a realistic test of the method. It was not clear whether this was the case. If the same cells are being mixed in different proportions then the experiment lacks the inter-individual variation that makes real applications of deconvolution harder. From the legend, at least the blood cells are from a single individual, and the target cell types may also be.

Indeed, the mixes shown in Figure 3 were generated by mixing different proportions of DNA from a specific target tissue (liver, lung, neurons and colon, each from one donor), with DNA from the blood of one healthy donor. This is now clarified in the text (legend to Figure 3). The purpose of this experiment was to to perform a first in vitro test of the deconvolution assay, following the in silico simulations shown in Figure 2, to assess the general performance of the assay and its accuracy in these well-defined conditions with minimal noise (note that the in silico simulations were in fact performed using data from multiple individuals). We agree that an additional set of in vitro mixing experiments, using samples from multiple donors, could have added information on accuracy of the procedure taking into account inter-individual variations in methylation, quantity of DNA preparation etc. However as pointed out by reviewer 1, even such a massive experiment would not be perfect as the sources of material are genomic DNA and not the real substrate of our work, namely cfDNA. We therefore decided to move, in this proof of concept study, straight to deconvolution of plasma samples from healthy individuals and patients with known pathologies, as shown in the next figures. We agree that for basic and clinical applications down the road, using an optimal platform (likely based on sequencing of a selected subset of targets, rather than arrays), more extensive mixing experiments, including multiple donors of blood and tissue, will be needed.

- The new Figure 3 should include the $x=y$ line.

We thank the reviewer for the detailed comments and have now added the requested reference lines (here and elsewhere).

- I don't feel the authors responded to the point in my original review regarding the bootstrapping method that was used to produce error bars of Figs 5c-e and Fig. 6c. The bootstrap procedure described will have the effect of assessing the robustness of their result to the choice of CpGs.

Following the reviewer comment, we have calculated the estimated errors of our algorithm using a parametric bootstrapping approach, where in each iteration we sample a new reference atlas with which we estimate the relative contribution of each cell type. Following Houseman et al (BMC Bioinformatics 2012, PMID 22568884), we used the original replicates for each tissue or cell type to estimate the mean CpG methylation and its standard deviation (for each CpG/cell type), which are then used to sample a methylation value. This new estimations had a very small effect on the error bars, and is now described in the revised Methods section.

- The issue concerning bootstrapping on CpGs also appears to arise in the new figures (3 and 7d)

Please see above. We thank the reviewer for the thoughtful and thorough comments.

Reviewer #1 (Remarks to the Author):

The authors have addressed all my comments.

I fully support publication of this manuscript.

To avoid ambiguity in units, authors should clarify in the newly added supplementary figures S5 and S6, if the y-axis shows diploid genome equivalents/ml, or haploid genome equivalents/ml.

The methods section should clarify how this cfDNA concentration was measured.

Reviewer #3 (Remarks to the Author):

I think the parametric bootstrap method the authors have implemented provides a better basis for assessment of the error in their inferences and therefore this aspect of the manuscript appears to have been improved.

The authors acknowledge that the in-vitro mixing experiments do not provide a reasonable assessment of the accuracy of their method (because they consist of mixtures from a single individual and lack the contribution to the difficulty of the problem arising from inter-individual variability). At the very least it should be highlighted in the manuscript that these results do not give an indication of how the method would be expected to perform in a realistic setting.

Detailed responses to reviewers' comments

Reviewer #1 (Remarks to the Author):

The authors have addressed all my comments. I fully support publication of this manuscript.

We thank the reviewer for the support and advice.

To avoid ambiguity in units, authors should clarify in the newly added supplementary figures S5 and S6, if the y-axis shows diploid genome equivalents/ml, or haploid genome equivalents/ml.

It is now clarified in the legends of these figures that the units refer to haploid genome equivalents/ml

The methods section should clarify how this cfDNA concentration was measured.

This has now been clarified in the methods section.

Reviewer #3 (Remarks to the Author):

I think the parametric bootstrap method the authors have implemented provides a better basis for assessment of the error in their inferences and therefore this aspect of the manuscript appears to have been improved.

We thank the reviewer for the support and advice.

The authors acknowledge that the in-vitro mixing experiments do not provide a reasonable assessment of the accuracy of their method (because they consist of mixtures from a single individual and lack the contribution to the difficulty of the problem arising from inter-individual variability). At the very least it should be highlighted in the manuscript that these results do not give an indication of how the method would be expected to perform in a realistic setting.

Thank you. We have added a statement along these lines in the in vitro section in the results.